Borneol promotes autophagic degradation of HIF-1α and enhances chemotherapy sensitivity in malignant glioma

Lin Luting 1
Luo Jingming 1
Wang Zeng wangzeng@zjcc.org.cn 2 3 4
Cai Xinjun zjtcmcxj@zcmu.edu.cn 1 5
1 School of Pharmaceutical Sciences, Zhejiang Chinese Medical University , Hangzhou , China
2 Zhejiang Cancer Hospital , Hangzhou , China
3 The Key Laboratory of Zhejiang Province for Aptamers and Theranostics, Hangzhou Institute of Medicine (HIM), Chinese Academy of Sciences , Hangzhou , China
4 Zhejiang Provincial Key Laboratory of Integrated Traditional Chinese and Western Medicine on Cancer, The Cancer Hospital of the University of Chinese Academy of Sciences , Hangzhou , China
5 Hangzhou Red Cross Hospital , Hangzhou , China
Sonowal Himangshu
Electronic publication date: 2024 Jan 3
Publication date: 2024
Volume: 12
Electronic Location ID: e16691
Received 2023 May 12; Accepted 2023 Nov 28
Copyright: ©2024 Lin et al.
Copyright year: 2024
Copyright holder: Lin et al.
License: This is an open access article distributed under the terms of the Creative Commons Attribution License, which permits unrestricted use, distribution, reproduction and adaptation in any medium and for any purpose provided that it is properly attributed. For attribution, the original author(s), title, publication source (PeerJ) and either DOI or URL of the article must be cited.
License URL: https://creativecommons.org/licenses/by/4.0/

Keywords: Borneol, TMZ, HIF-1α, Autophagy, Glioma

Funding: The Natural Science Foundation of China 82003669 Natural Science Foundation of Zhejiang Province LYY18H300001 Chinese Medicine Science and Technology Plan of Zhejiang Province 2021ZB213 2021ZB036 Zhejiang Medical and Health Science and Technology Plan 2020KY741 This work was supported by the Natural Science Foundation of China (grant number 82003669), the Natural Science Foundation of Zhejiang Province (grant number LYY18H300001), the Chinese Medicine Science and Technology Plan of Zhejiang Province (grant number 2021ZB213, 2021ZB036), and the Zhejiang Medical and Health Science and Technology Plan (grant number 2020KY741). The funders had no role in study design, data collection and analysis, decision to publish, or preparation of the manuscript.

==============================
Background

Gliomas are characterized by high mortality rates and resistance. Even with conventional chemotherapy the prognosis of glioblastoma remains poor. Many medications are not optimally effective due to limited bioavailability. The bioavailability of medicine can be enhanced by borneol, a monoterpenoid substance. In this study, we investigated the effect of borneol, a commonly used Chinese medicine, on chemosensitivity in C6 glioma and U251 human glioma cell lines and elucidated its therapeutic molecular targets.

Methods

The chemosensitivity-inducing effects of borneol in C6 and U251 cells were examined using CCK8 and clonal formation assays. The mechanism underlying the effect of borneol was evaluated through immunohistochemistry and western blotting assays. Further, the number of autophagosomes was determined via transmission electron microscopy. Finally, the chemical sensitization effect of borneol was evaluated in SD rats after C6 orthotopic tumor transplantation.

Results

Borneol increased cytotoxicity in C6 and U251 cells in response to temozolomide (TMZ). In addition, through transmission electron microscopy, western blotting, and immunohistochemical tests, we found that borneol combined with TMZ significantly increased the level of autophagy and that hypoxia inducible factor-1(HIF-1α) is a candidate target through which borneol enhances the cytotoxic effect of TMZ. Borneol’s ability to enhance HIF-1α degradation was counteracted following the administration of autophagy inhibitors. In vivo, borneol treatment was found to enhance the anticancer effect of TMZ and delay tumor progression, and this effect was closely related to its ability to promote the autophagic degradation of HIF-1α.

Conclusions

HIF-1α might be a valid therapeutic target of borneol, which can be potentially applied as a chemosensitizing drug used for glioma treatment.

Introduction

Gliomas are malignant tumors of glial cell origin that are highly infiltrative and invasive. Glioblastoma multiforme (GBM) accounts for approximately 57% of all gliomas and is the most common malignant brain tumor, accounting for 48% of all primary malignant central nervous system tumors (Tan et al., 2020). Despite the recent development of multimodal treatments for glioma, including surgery, chemotherapy, radiotherapy, targeted therapy, and supportive care, its overall prognosis remains poor, with unsatisfactory long-term survival and a 5-year survival rate of only 5.8% (Tan et al., 2020). In addition, drug resistance and poor dosing are major barriers to the use of temozolomide (TMZ). Emerging evidence suggests the benefits of combination therapy using chemotherapeutic agents along with natural products. Natural products, as adjuvant anticancer drugs, are an acceptable therapeutic approach, owing to their accessibility, suitability, and reduced cytotoxicity (Hashem et al., 2022). Natural products could also increase efficacy and reduce side effects when exploited in combination with chemotherapeutic drugs.

Borneol is commonly used in traditional Chinese medicine for its effects of waking up the brain, opening the body, relieving heat and pain, antisepsis, and muscle building (Committee, 2020). Further, it has been used as an adjuvant drug in combination with other drugs to enable effective treatment at lower doses with fewer side effects (Liu et al., 2021). Recently, borneol was found to enhance the antitumor effects of doxorubicin (Cao et al., 2019), bisdemethoxycurcumin (Jianping Chen, Su & Chen, 2014), curcumin (Chen et al., 2015b), and cisplatin (Li et al., 2022a). Previous studies have also proven that borneol can improve radiotherapy efficacy in glioma, and its mechanism might be related to the downregulation of hypoxia inducible factor-1 (HIF-1α) expression (Wang et al., 2020b; Qinglin et al., 2021).

The expression level of HIF-1α is closely related to tumor cell proliferation, differentiation, apoptosis, phenotype determination, angiogenesis, energy metabolism, and resistance to therapy (Domènech et al., 2021). The Cancer Genome Atlas data has shown that its expression level is higher in glioblastoma tissues than in normal tissues, and high expression of HIF-1α was found to result in shorter disease-free survival and overall survival (Wang et al., 2020a). Downregulation of its expression is of great significance for improving antitumor efficacy (Semenza, 2013).

Previous studies have found that borneol promotes cellular autophagy via the mTORC1/eIF4E/HIF-1α axis, thereby sensitizing glioma cells to radiation (Qinglin et al., 2021). However, few reports exist pertaining to the effect of borneol on, and the role of HIF-1α in the chemosensitivity of gliomas. Therefore, the purpose of this study was to assess the effect of borneol on glioma cells chemosensitization and to investigate the role of autophagy in this process. In this study, we used C6 rat glioma cells and a Sprague–Dawley (SD) rat transplanted glioma model to evaluate the effect of the combination of borneol and TMZ on the growth of C6 and U251 glioma cells in vitro and in vivo, as well as the mechanism underlying such effects.

Materials & Methods

Chemicals

Borneol was obtained from Shanghai Yuanye Biotechnology Co., Ltd. (Shanghai, China), and Dulbecco’s Modified Eagle Medium (DMEM) was purchased from ThermoFisher Biochemical Products Co., Ltd. (Gibco, Waltham, MA, United States). Temozolomide and Chloroquine (CQ) was obtained from Selleck Co., Ltd. (Houston, Texas, United States). TdT In Situ Apoptosis Detection Kit - Fluorescein was purchased from R&D Systems, Inc. (Minneapolis, MN, United States). Cell Signaling Technology (Danvers, MA, USA), Affinity Biosciences (Jiangsu, China), ZEN-BIOSCIENCE (Chengdu, China) and MULTISCIENCESL (Hangzhou, China) supplied the antibodies. A CCK8 Kit was purchased from Shanghai Life iLab Biotech Co., Ltd. (Shanghai, China).

Cell proliferation test

C6 cells were obtained from the Laboratory of Traditional Chinese Medicine Preparations, Zhejiang Chinese Medical University. U251 cells were obtained from Wenzhou Medical University. U251 and C6 cells were cultured at 37 °C in a 5% CO2 incubator in DMEM supplemented with 10% FBS, as well as penicillin and streptomycin (100 U/mL). Cells were seeded in 96-well plates and treated for 48 h with 0–800 µM TMZ and/or 0–80 µg/mL borneol. The cell viability of U251 and C6 cells was determined using CCK8 assay. Cell proliferation was analyzed by measuring the absorbance at 450 nm using a multifunctional enzyme marking instrument (Thermo Scientific™ Varioskan™ LUX; Thero Fisher Scientific, Waltham, MA, USA). The cell activity decreased considerably in a concentration-dependent manner (Fig. 1A). For C6 cells, when the TMZ concentration exceeded 250 µM, the decline in survival rate slowed remarkably. Similarly, for U251 cells, the IC50 reached approximately 200 µM (Fig. 1A). Additionally, based on the results of joint drug screening using a gradient of TMZ and borneol (Fig. 1C), the cell viability decreased considerably when the borneol concentration was 40 µg/mL. Therefore, we determined that combined treatment concentrations would involve borneol concentrations of 40 µg/mL, and TMZ concentrations of 200 µM (U251), and 250 µM (C6).

Figure 1 Borneol (Bor) combined with temozolomide (TMZ) shows enhanced cytotoxicity against glioblastoma.

(A) A CCK8 viability assay revealed the effect of TMZ treatment alone for 48 h on C6/U251 cells (n = 5). (B) Effects of borneol for 48 h on cell activity of C6/U251 alone (n = 5). (C) Effect of TMZ combined with borneol treatment for 48 h on the activity of C6/U251 cells (n = 5). For the replicate numbers, they are biological replicates. (D) Clonogenic assay with borneol and TMZ treatment for 7 d, based on staining, using C6 and U251 cells (n = 3 independent experiments). In the figure, “T” refers to TMZ in µM and “B” represents borneol in µg/mL. Each value represents means ± SEM. A comparison between the control group (*P < 0.05, ***P < 0.01, ****P < 0.0001) is shown.

Clonogenic assay

One thousand cells were added into each well and incubated in a CO2 incubator. After 1 week of incubation, when visible colonies formed, the 6-well plates were removed, the culture medium was discarded, and the wells were rinsed once with phosphate-buffered saline (PBS). Methanol was added to each well, and the plate was fixed for 30 min. After discarding the methanol, 0.1% crystal violet (Solarbio, Beijing, China) was added and the cells were stained for 10 min, then washed.

Western blotting (WB)

Cell culture medium was collected, cells were lysed, total protein was extracted, and protein was separated using 8% and 12% SDS-PAGE. The proteins were then transferred to a polyvinylidene difluoride (PVDF) membrane, and blocked with 5% skim milk for 1.5 h at room temperatureat. Blots were incubated with the primary antibodies overnight, washed, incubated with secondary antibodies for 1.5 h, and washed again. Enhanced chemiluminescence (ECL) was used for color development and exposure. Bands detected using WB were scanned in grayscale mode using a multifunction imager (Thermo Fisher Scientific, Waltham, MA, USA). The protein expression of HIF-1α, Beclin-1, and LC3 was then detected. Finally, ImageJ software was used to quantify the WB bands. The antibodies used in this experiment were as follows: HIF-1α (1:1000; Affinity), Beclin-1 (1:1000; Affinity), LC3A/B (1:1000; CST), and GAPDH (1:3000; ZENBIO), Goat Anti-Rabbit IgG(H+L) HRP (1:20000; MULTISCIENCES)

In vivo antitumor activity

SPF-grade male SD rats were obtained from Zhejiang Chinese Medical University Laboratory Animal Research Center. All animal experiments were performed in compliance with guidance of the Zhejiang Chinese Medical University Animal Care and Use Committee (IACUC-20211018-14). After 2 weeks of acclimation, animals were inoculated with C6 cells (1.0 × 106 cells for each rat) using the brain stereotaxic method to establish a glioma model (Zhang et al., 2011). One week after modeling, rats were randomly divided into four groups as follows: control, borneol (16 mg/kg), TMZ (50 mg/kg), and combined treatment group (six rats each). Rats in each group were intragastrically administered the drug daily for nine consecutive days. The general condition of rats as well as the weight of tumors were recorded, and Hematoxylin-Eosin (HE) staining was used for identification.

Transmission electron microscopy (TEM)

Rat brain glioma tissues were first fixed with glutaraldehyde, rinsed in buffer, fixed with 1% osmium acid, washed, and dehydrated in an alcohol gradient, soaked, embedded, polymerized, stained based on sections, and observed via TEM (Hitachi-7650).

Immunohistochemistry (IHC)

Rat brain glioma tissues samples were fixed in paraformaldehyde, dehydrated, and embedded in paraffin. Sections were dewaxed and hydrated, repaired in sodium citrate buffer for 4 h (60 °C oven), inactivated by 3% H2O2 for 15 min, washed with distilled water, and incubated with primary antibodies overnight at 4 °C. The sections were then washed, incubated with secondary antibodies, and rinsed with PBS. Diaminobenzidine was used for development, and the slides were rinsed with tap water, stained with hematoxylin, and sealed with a neutral resin. For light microscopy (Leica microscope; Leica, Wetzlar, Germany), the areas and positively stained cells were counted in six random non-overlapping fields of view at 400 × magnification.

TUNEL assay

Rat brain glioma tissues were fixed in paraformaldehyde, dehydrated, and embedded in paraffin. The sections were then dewaxed and hydrated. The samples were covered with 50 µL of Proteinase K Solution and incubated at 37 °C for 30 min. These were then washed twice with distilled water. Next, the slides were immersed in 1X TdT Labeling Buffer for 5 min. The samples was covered with Labeling Reaction Mix and incubated at 37 °C for 1 h in a humidity chamber. To stop the labeling reaction, the samples were immersed in 1X TdT Stop Buffer for 5 min at room temperature. These were then washed twice with PBS for 5 min each at room temperature. Next, the samples were covered with Strep-Fluorescein Solution and incubated for 20 min at room temperature in the dark. After washing with PBS, the slides were sealed with DAPI-containing mounting tablets. The samples were observed and images were captured using a confocal microscopy (Leica confocal microscopy).

Statistical analysis

GraphPad 8 was used for statistical analysis. The number of animals used for the in vivo models in this study was determined based on previous calculations, and the in vitro assays were repeated at least three times. Data are expressed as the mean ± SEM (x ± s), and differences between groups were compared using the t-test. Statistical significance was set at P < 0.05.

Results

Borneol treatment enhances the efficacy of TMZ in glioma

Cell viability was assessed based on a CCK8 assay. C6 and U251 cells were treated with varying doses of TMZ for 48 h. The cell viability assay decreased significantly in a concentration-dependent manner (Fig. 1A). Moreover, U251 cells were more sensitive to TMZ than C6 cells. For C6 cells, when the TMZ concentration was higher than 250 µM, the decline in the cell viability rate slowed significantly, whereas for U251, when the concentration was 200 µM, the viability almost reached the IC50. As shown in Fig. 1B, borneol treatment alone mildly inhibited the growth of C6 cells and U251 cells in the concentration range of 10 µg/mL to 80 µg/mL. In addition, for C6 cells, a significant difference was observed based on borneol concentration up to 40 µg/mL compared to that in the controls (P < 0.05). The results of drug concentration screening, based on TMZ and borneol gradient, showed that the activity of C6 cells decreased remarkably under the combined treatment of 250 µM TMZ and 40 ug/mL borneol (Fig. 1C), compared with other borneol concentrations. For U251 cells treated with 200 µM TMZ, the activity of the cells showed a relatively large decreasing trend when combined with 40 ug/mL borneol, compared with other borneol concentrations.

Subsequently, the effect of borneol combined with TMZ on colony formation was measured in C6 and U251 cells. The results showed that colony formation was reduced. Cell death was increased in the combined treatment group compared to those in the group treated with the same concentration of TMZ alone (Fig. 1D). However, treatment with borneol alone had no significant effect on colony formation. In conclusion, these results suggest that combined treatment with borneol can enhance the cytotoxicity of TMZ in C6 and U251 cells and inhibit cell proliferation.

Effects of borneol and/or TMZ on C6 and U251 protein expression in C6 and U251 cells

Next, we assessed the effectiveness of borneol treatment combined with TMZ, based on protein expression in C6 and U251 cells, compared to that with TMZ treatment alone. We observed the effects of borneol treatment and TMZ on HIF-1α and autophagy levels by WB experiments. The concentration of autophagy inhibitor (CQ) was determined to be 5 µM (C6) and 10 µM (U251) through cytotoxicity tests (Fig. 2C). Beclin-1 and LC3, as markers, are involved in autophagy. Beclin-1 is also a tumor suppressor, and its expression level is negatively correlated with the activity of tumor cells. Our results showed that both borneol and TMZ alone could reduce HIF-1α expression, whereas their combination further reduced HIF-1α protein expression (Figs. 2A and 2B), consistent with our previous results (Qinglin et al., 2021). Moreover, after the use of autophagy inhibitors, the effect of borneol on promoting HIF-1α degradation was reversed (Figs. 2D–2E). From the perspective of autophagy, compared with those in the control group, Beclin-1 levels were increased upon single treatment, and combined treatment also significantly promoted its expression in U251 cells compared to that with TMZ alone (Figs. 3D and 3E). Meanwhile, in C6 cells, combined treatment tended to increase Beclin-1 expression, and the expression level of Beclin-1 increased as the concentration of borneol increased. However, no significant difference (P = 0.0699) was found between the combined treatment and TMZ treatment alone (Figs. 3A and 3B). Further, the results of our study indicated that the level of LC3 II/I was significantly increased by combined treatment when compared with that in the control group, and combined treatment tended to further increase the level of LC3 II/I compared to that with TMZ treatment alone (Figs. 3C and 3F). In general, our results proved that borneol and TMZ alone could promote autophagy in C6 and U251 cells, as well as HIF-1α protein degradation. The combination of the two further increased the level of autophagy and promoted the degradation of HIF-1α.

Figure 2 Expression levels of HIF-1α in C6 and U251 glioma cells.

Western blots and relative statistical analysis of HIF-1α in C6 cells (A) and U251 (B) cells after borneol and temozolomide (TMZ) treatment for 48 h (n = 3). (C) Cell viability was measured by CCK8 assay on treatment with 0-10 µM CQ for 48 h (n = 5 biological replicates). Protein expression and relative statistical analysis of HIF-1α in C6 cells (D) and U251 (E) cells after borneol and Chloroquine (CQ) treatment for 48 h (C6, n = 3; U251, n = 4). Each value represents means ± SEM. A comparison between the control group (*P < 0.05, **P < 0.01, ***P < 0.001) and the TMZ/CQ group (▴P < 0.05) is shown (n = 3). For the replicate numbers of western blotting assays, they are independent experiments.

Figure 3 Expression levels of Beclin-1 and LC3A/B in C6 and U251 glioma cells after borneol and temozolomide (TMZ) treatment.

(A–C) protein expression of Beclin-1 and LC3A/B in C6 cells and their relative statistical analysis. (D–F) protein expression of Beclin-1 and LC3A/B in U251 cells and their relative statistical analysis. Each value represents means ± SEM. A comparison between the control group (*P < 0.05, **P < 0.01, ***P < 0.001, ****P < 0.0001) and the TMZ group (▴P < 0.05) is shown (n = 3). For the replicate numbers, they are independent experiments.

Enhanced in vivo anticancer efficiency of TMZ when combined with borneol

Subsequently, an in-situ transplantation model of C6 rat glioma was generated using brain stereolocalization to evaluate the efficacy of borneol in combination with TMZ for the treatment of glioma. Our results suggest that both borneol and TMZ alone could reduce the tumor weight when compared to that in the control group, and that combination treatment inhibited tumors more effectively than TMZ alone, consistent with the results of our in vitro experiments (Fig. 3A). Based on the analysis of body weight of SD rats at the beginning of modeling and during administration (from the 7th day onward), the glioma brain model was successfully created approximately one week after modeling, but there was no significant difference in the influence of different treatments on the body weight of rats (Fig. 4B). Moreover, HE staining revealed that the tumor density of the borneol/TMZ combined treatment was significantly lower compared to that with in separate treatments, indicating that the combined treatment could further inhibit the cell proliferation (Fig. 4D). In addition, TEM revealed that both borneol and TMZ could promote autophagy and that combined borneol /TMZ treatment further stimulated autophagy (Fig. 4C). In conclusion, borneol induces autophagy in glioma cells and enhances the tumor-suppressive effects of TMZ.

Figure 4 Borneol (Bor) enhances the anticancer efficiency of temozolomide (TMZ) in vivo.

(A) Brain glioma weights after treating rats with borneol (Bor) and TMZ (n = 6). (B) Changes in body weight of SD rats with in situ glioma after treatment with borneol and TMZ (the drug was administered on the seventh day after molding, n = 6). Each value represents means ± SEM. (C) Glioma tissues viewed via transmission electron microscopy; the red arrows represent autophagosomes (TEM; 25,000×). (D) HE staining of rat brain gliomas after treatment with borneol and TMZ. A comparison between the control group (*P < 0.05) and the TMZ group (▴P < 0.05) is shown. For the replicate numbers, they are biological replicates.

IHC evaluation of the effects of borneol and/or TMZ on protein expression in tumor tissue

Figure 5 shows, the immunohistochemical analysis of the tumor tissues. Compared with that in the untreated-tumor bearing group, the level of HIF-1α expression in animal glioma tissues treated with borneol and/or TMZ was significantly reduced (Fig. 5A), and the effect of combined therapy was even greater. In addition, the combined use of borneol and TMZ significantly increased the expression levels of Beclin-1 and LC3A/B in glioma tissues compared to those in the untreated controls (Figs. 5B and 5C). Consistent with this, TEM inspection (Fig. 5B) revealed that compared with that in the untreated control group, a large number of autophagosomes was observed in the borneol, TMZ, and combined treatment groups, and compared to that with TMZ alone, the combined treatment further promoted autophagy levels. In accordance with previous studies, borneol inhibited HIF-1α expression by regulating the mTORC1/eIF4E pathway (Wang et al., 2020b), and we found that borneol might make glioma cells sensitive to TMZ by promoting HIF-1α autophagic degradation.

Figure 5 Immunohistochemical detection of HIF-1α, Beclin-1, and LC3A/B expression levels after treating gliomas with borneol (Bor) and temozolomide (TMZ).

Immunohistochemical detection of (A) HIF-1α, (B) Beclin-1, and (C) LC3A/B. Each value represents means ±SEM. A comparison between the control group (**P < 0.01, ***P < 0.001, ****P < 0.0001) and the TMZ group (▴P < 0.05, ▴▴▴P < 0.001, ▴▴▴▴P < 0.0001) is shown (n = 6, 400×). For the replicate numbers, they are biological replicates.

Effects of borneol and TMZ on apoptosis of tumor cells

We evaluated the effect of borneol on apoptosis by conducting a TUNEL (Fig. 6). Compared to the control group, treatment with borneol alone showed a slightly increase in the number of apoptotic cells. Both TMZ treatment alone and the combined treatment significantly increased apoptosis level. Furthermore, the combined treatment showed a further increase in apoptosis compared to TMZ treatment alone. These experimental results are consistent with those of previous studies (Wang et al., 2020b). Borneol alone can down-regulate HIF-1α and promote apoptosis(Wang et al., 2020b), which further validates our study’s findings that borneol induces apoptosis in glioma cells by promoting autophagy to degrade HIF-1α.

Figure 6 Apoptosis detection in C6 glioma cells.

The apoptosis of brain glioma cells after treatment was observed by immunofluorescence. (n = 5, 200×). For the replicate numbers, they are biological replicates.

Discussion

High-grade gliomas with an incomplete envelope and slow growth without obvious borders and with normal brain tissue are common intracranial malignant tumors (Jiang et al., 2021). Therefore, complete resection through surgery is difficult, and TMZ-based postoperative chemotherapy is the primary treatment for this disease. However, in clinical practice, achieving the expected effect with postoperative chemotherapy is difficult, mainly because of the frequent occurrence of tumor cell tolerance to chemotherapeutic drugs, which severely limits treatment success in patients with advanced glioma. Therefore, the development of efficient chemical sensitizers with low toxicity is important to enhance the therapeutic effects of chemotherapeutic drugs and reduce dose-related adverse effects. Our research showed that the borneol and TMZ combination treatment exhibited stronger cytotoxicity than TMZ alone. In vivo, this combined treatment further decreased the tumor weight and volume.

There have been many studies on borneol promoting drug penetration through various physiological barriers, including blood–brain barrier, skin, and mucosa (Zhang, Fu & Zhang, 2017; Zhang et al., 2019; He, Shen & Li, 2011). In recent years, borneol has been found to have potential as a chemotherapy sensitizer (Lai et al., 2020). Cao et al. (2019) found that borneol not only significantly enhanced intracellular uptake of DOX by cells, but also enhanced DOX-mediated DNA damage by activating ROS overproduction, resulting in G2/M phase arrest of damaged cells and eventual inhibition of cell growth. Meanwhile, Chen et al. (2015a) also found that borneol could enhance the uptake of demethoxycurcumin by cells, and promote the anti-cancer effect of demethoxycurcumin by regulating ROS-mediated DNA damage. In addition, studies have also shown that borneol plays an anticancer role by promoting the expression of pro-apoptotic MAPK family members and PI3K/Akt pathway proteins (Li et al., 2021; Wen-qiang Cao et al., 2020). Our previous studies found that borneol can promote apoptosis by down-regulating HIF-1α expression (Wang et al., 2020b), and stimulate autophagy by targeting the mTORC1/eIF4E signaling pathway, thereby down-regulating HIF-1α expression and rendering glioma cells sensitive to radiotherapy (Qinglin et al., 2021). However, the mechanism by which borneol sensitizes TMZ-based chemotherapy remains elusive and the role of HIF-1α and autophagy in this process remains unpredictable.

In this study, the combination of borneol and temozolomide was screened to verify that the combination had stronger cytotoxicity than that of TMZ alone. At the same time, WB results suggested that combined administration further down-regulates HIF-1α expression, and after the use of autophagy inhibitors, the down-regulated HIF-1α protein expression of borneol is reversed, suggesting that borneol may degrade HIF-1α by promoting autophagy. Subsequently, the expression level of autophagy-related proteins was detected, and the results showed that both borneol and TMZ could increase the level of autophagy, and the combined treatment further promoted autophagy. Consistent with this, Li et al. (2022b) found that d-borneol increases the sensitivity to cisplatin by activating autophagy.

The HIF-1α plays a key role in cancer cell metastasis and drug resistance, which is an important factor limiting the response of many cancers to chemotherapy (Dong et al., 2022; Joo-Yun Byun et al, 2022). Lo Dico et al. (2018) found that the downregulation of HIF-1α expression increases the response of TMZ-resistant cells. However, chaperone-mediated autophagy inhibition could induce TMZ resistance in TMZ-sensitive cells. In addition, Monika E. Hegi et al. (2005) found that the effectiveness of TMZ depends on silencing the MGMT gene. However, in approximately 55% of patients, MGMT is not silenced; therefore, they do not benefit from TMZ. In GBM stem cells, the downregulation of HIF-1α expression reduces MGMT expression, thereby increasing the TMZ response in TMZ-resistant cells, which suggests that combining this inhibitor with TMZ might be beneficial (Persano et al., 2012; Cowman & Koh, 2022), which is consistent with our results. Meanwhile, the study conducted by Dong et al. (2022) revealed that the down-regulate HIF-1α expression, either through a knockdown or pharmacological inhibition, can effectively inhibit glycolysis and reverse 5-FU resistance. As HIF-1α serves as a key upstream-regulator of the entire glycolytic pathway, this approach may prove more effective compared to other agents (Dong et al., 2022). These findings are consistent with our results, further supporting our conclusions.

Notably, in some cases, TMZ-induced autophagy can inhibit apoptosis and play a protective role in autophagy (Zanotto-Filho et al., 2015). However, in other cases, autophagy can induce apoptosis and the death of glioblastoma cells after TMZ treatment. Promoting autophagy might further enhance the effects of chemotherapy (Li et al., 2022b). The ability to regulate autophagy in response to TMZ treatment could hold great promise for preventing chemotherapy resistance and improving anticancer therapy efficacy. Therefore, inhibiting HIF-1α by controlling autophagy is a potential therapeutic strategy. The in vitro and in vivo experimental results of our study also proved that borneol combined with TMZ can further improve the level of autophagy, and that this combination can differentially regulate various pathways to promote autophagy. Our previous studies showed that borneol-induced autophagy is likely mediated by the mTORC1/eIF4E pathway (Qinglin et al., 2021), which is different from the autophagy pathway induced by TMZ. However, further research is needed to understand the molecular mechanisms underlying autophagy, because the network associated with autophagy might be complex.

Conclusions

Here, our study showed that borneol can improve TMZ chemotherapy sensitivity, and that HIF-1α is a potential target for improving the antitumor efficacy of TMZ, which is closely related to the promotion of the autophagic degradation of HIF-1α, thus improving chemotherapy sensitivity in glioma cells. The combination of TMZ and borneol, a “guide” drug used in Chinese medicine, provides a promising approach to the treatment of glioma.

Supplemental Information

Supplemental Information 1 Raw data

Click here for additional data file.

Supplemental Information 2 Original western blot for three repeats

Click here for additional data file.

Supplemental Information 3 CCK8; tumor weight; rat weight and AOD of IHC data

Click here for additional data file.

This work was done at the Laboratory of Hangzhou Integrative Medicine Hospital Affiliated to Zhejiang Chinese Medical University (Hangzhou Red Cross Hospital), China. The authors are grateful to the staff in the laboratory for their technical assistance.

Additional Information and Declarations

Competing Interests

Author Contributions

Animal Ethics

Data Availability

The authors declare there are no competing interests.

Luting Lin conceived and designed the experiments, performed the experiments, analyzed the data, prepared figures and/or tables, and approved the final draft.

Jingming Luo performed the experiments, prepared figures and/or tables, and approved the final draft.

Zeng Wang conceived and designed the experiments, authored or reviewed drafts of the article, and approved the final draft.

Xinjun Cai conceived and designed the experiments, authored or reviewed drafts of the article, and approved the final draft.

The following information was supplied relating to ethical approvals (i.e., approving body and any reference numbers):

Zhejiang Chinese Medical University Laboratory Animal Research Center provided full approval for this research (IACUC-20211018-14)

The following information was supplied regarding data availability:

The raw WB pictures and raw data are available in the Supplementary Files.

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
