# Peer review of "Borneol promotes autophagic degradation of HIF-1α and enhances chemotherapy sensitivity in malignant glioma"

_PeerJ, doi:10.7717/peerj.16691_

## Round 0.1 · original submission · Major Revisions

Dear Dr. Cai,
We have received an extensive and elaborate review report from a panel of expert reviewers for your manuscript titled," Borneol promotes autophagic degradation of HIF-1α and enhances chemotherapy sensitivity in malignant glioma". I welcome you to revise and resubmit the manuscript and address the points raised by the reviewers.

Thank you for submitting your manuscript to PeerJ.

With kind regards,
Himangshu Sonowal
Academic Editor

Reviewer 1 ·

Basic reporting

The manuscript deals with borneol promotes autophagic degradation of HIF-1 alpha and enhances chemotherapy sensitivity in malignant glioma. To confirm this hypothesis, authors performed various experiments. The plan and experimental setup is good, but still certain aspects need to be improved.

Experimental design

In materials and methods should be even in details information. Example – In cell proliferation assay, authors mentioned that proliferation measured by measuring absorbance at 450 using which instrument with company name and country. In western blot, line number 96-96: protein was separated not isolated. What is the blocking agent used with how many percentages? Also, authors need to mention the primary and secondary antibody company and antibody dilution. How blots were imaged? Using which instrument? Which imaging analysis tool used to quantify the western blot bands?

In vivo antitumor activity assay – Establishment of glioma model should be in detail or they should cite publication.

In IHC, line no 119, the antigen is not repaired, its retrieval. Antigen retrieval method should be in details. Microscope company?

In IHC figure 2 and 4 were unbale to see the difference between the groups. It looks like not developed completely.

As authors mentioned, they have used Student Ttest, but here groups are more than two which means, they should use One way- ANOVA. In particular, if they use only two groups to compare then they can use Student- ttest.

Scientific notation should be used correctly like example symbols, in vivo and invitro should be in italics.

Abbrevation should be expanded if they use for first time example HIF-1α

Validity of the findings

1. Authors claimed that borneol promotes autophagic degradation, but how they confirm that apoptosis not involves.
2. Also, it’s important to use inhibitors for autophagy (Eg. Bafilomycin).

Additional comments

Manuscript English need to be improved.

Reviewer 2 ·

Basic reporting

The manuscript is overall comprehensible, but it could benefit from additional language editing to enhance its readability and clarity. While the introduction introduces sufficient background, there are instances where key information lacks supporting references. I recommend the authors revise these areas, ensuring every substantial piece of information or claim is properly cited. There are also some specific points for potential improvements in basic reporting. See point-by-point comments below.

- Line 44-46: "Emerging evidence suggests the benefits of combination therapy using anticancer together with natural products (as anticancer drugs) as an acceptable therapeutic approach, owing to their accessibility, suitability, and reduced cytotoxicity [2]." – This sentence appears a bit convoluted. Please check and revise.
- Line 50-51: "Borneol is commonly used in traditional Chinese medicine for its effects of waking up the brain, opening the body, relieving heat and pain, antisepsis, and muscle building." – please supplement with appropriate references.
- In the figure captions, please indicate more detailed information about the data, including the number of replicates, what the error bars represent, treatment duration, etc.
- Cell viability figures: although the notation used is '(%)', the values seem to be normalized to 1.0 rather than presented as percentages. Please check and revise accordingly.
- More justification is needed for the concentrations chosen for both Temozolomide and Borneol in the experiments. The concentration of 250 µM for TMZ and 80 µg/mL for Borneol seem high. It would be beneficial to provide evidence that such concentrations can be realistically achieved in plasma during in vivo applications. See additional comments in the Validity of Findings section.
- It seems there might be a discrepancy between the figures and the text with respect to Figures 2 and 3. Please verify if the descriptions in the main text correspond accurately to the uploaded figures.
- Figure 2B: clarify what the red arrows represent in the Figure caption.
- Line 154-157: "Past research has found that borneol can significantly improve autophagy levels in human primary glioma cells, enhance radiotherapy sensitivity, and improve radiotherapy efficacy by promoting HIF-1a degradation." – please provide a reference. It is also recommended to move this to the introduction or discussion section instead.
- Line 141-142: "Survival analysis revealed that the combination of TMZ and borneol exerted a synergistic effect." The experiment being described appears to be a viability assay. The term 'survival analysis,' however, typically refers to a statistical analysis often used in clinical trials to measure the proportion of individuals surviving over time under different conditions. Please ensure the correct terminology is used accurately. Also, the synergistic effects could not be justified - see additional comments in the Validity of Findings section.
- Lines 218-219: "A prior study showed that HIF-1α might also be a key factor involved in the sensitizing effects of borneol on glioma cells". Please include relevant references. Furthermore, the term 'sensitizing effects' lacks specificity – a bit more detail would help.
- The labeling scheme for the graphs and Western blots could benefit from enhancements to improve readability and comprehension. Specifically, incorporating a two-row labeling structure might help. For instance, in the first row: 'Borneol (µg/mL): 0, 10, 80, … [additional concentrations]', and below it, in the second row: 'TMZ (µM): [respective concentrations]'. This revised labeling format would aid in preventing potential confusion and misinterpretations.
- Figure 4 IHC AOD raw data was not provided.

Experimental design

- Western blot quantifications were done extensively, but it was not clearly described how the quantifications were done. Please indicate the software used and if any background subtraction was conducted.
- Further elaboration is needed in the methods section to ensure the clarity and reproducibility of the experiments. Specifically, details about the equipment used, such as the model and manufacturer of the Transmission Electron Microscope (TEM), should be provided. In addition, catalog numbers of all reagents, including antibodies, should be included to allow accurate replication of the experiments. Furthermore, crucial details about the timing and dosing frequency of treatments, such as the duration of rat treatment and the frequency of drug administration, need to be clarified. Similarly, the method of drug administration should be clearly stated. Please revise the methods section to address these points.
- Synergy could not be robustly justified by the existing experimental approach. Please consider a drug combination screen and synergy scoring. See additional comments in the Validity of Findings section.
- There is an absence of an experiment to link autophagy initiation and autophagy degradation HIF-1α. The authors could consider an autophagy inhibitor experiment to monitor changes to HIF-1α degradation by Borneol treatment. See additional comments in the Validity of Findings section.

Validity of the findings

There are some significant limitations to the validity of some findings. Some comments are provided below:
- As previously mentioned in Basic Reporting, the clinical relevance of the concentrations of the drugs used needs to be evaluated. Concerns about Temozolomide concentrations used in in vitro studies are raised in this article by Poon et al. (PMID: 34794398). The authors should provide a more detailed discussion in this manuscript as it would affect the interpretation and clinical relevance of the results.
- It appears that the health of the rats was monitored, but the weight of the rats (at the endpoint or over time) was not reported in this manuscript. If there is data on the weight of the animal, it may help to include it. Given the drugs used and the dosage, it's important to monitor the health of the rats, as this could impact tumor growth. The observed effect on tumor sizes is meaningful only if it is not due to the compound's impact on the general health of the animals. Additionally, I recommend including representative images (if available) of the tumors to provide visual confirmation of the observed effects.
- Line 141-142: "combination of TMZ and borneol exerted a synergistic effect" is not justified. The differences described in Figure 1 are not readily discernible, making it challenging to ascertain whether the observed effects of Borneol and TMZ treatments are merely additive or if there's genuine synergy. It may be beneficial to calculate IC50 curves for TMZ both with and without a fixed concentration of Borneol. Changes in the IC50 could provide a clearer indication of whether the combined treatment results in enhanced efficacy. Additionally, a drug combination screen in cells with a gradient of TMZ and Borneol is recommended. This would be able to tell you about a synergy score, which could help to justify synergy.
- In the western blot experiments reported in Figure 3A-B: the evidence presented does not convincingly demonstrate the suggested sensitization. The effects observed could possibly be attributed to independent, additive effects rather than synergistic interactions. Please more appropriately control the experiments. It may be better to constrain one concentration of Borneol and vary the concentration of TMZ to understand any shift in the concentration required to achieve the HIF-1α reduction.
- While the observed upregulation of Beclin-1 and LC3A/B indeed signals the initiation of autophagy, it does not necessarily justify that "Borneol promotes autophagic degradation of HIF-1α". The authors did mention in lines 154-157 that previous research has shown this effect but did not provide a reference. The degradation of HIF-1α may result from other pathways affected by Borneol, not just autophagy. Therefore, to provide better support for this claim, it would be useful to conduct further experiments, such as inhibiting autophagy with a specific inhibitor and then monitoring HIF-1α levels like in Figure 3A, to see if there may be inhibition of HIF-1α degradation.

Additional comments

The current version of the manuscript contains substantial limitations, necessitating a comprehensive revision and additional data. As it stands, the central argument that Borneol sensitizes cells to Temozolomide is not convincingly supported by the presented data. More robust and detailed experimental evidence is needed to convincingly demonstrate this point. Comments and suggestions are provided in their respective sections.

Reviewer 3 ·

Basic reporting

The manuscript (#85450) entitled, “Borneol promotes autophagic degradation of HIF-1α and enhances chemotherapy sensitivity in malignant glioma” reports the synergistic effect of Borneal in enhancing anticancer effect of TMZ in glioma and tried to validate its mechanistic role via HIF1 mediated autophagy of cancer cells both in vitro and in vivo. The study design and methodology adopted is good and is as per standards. Technically, the effect of borneol on glioma regression was well demonstrated. The results are documents in good quality representative images and graphs. The manuscript is well written. The language and grammar are appreciable. The outcome of the study is simple and straight proving that Borneol can induce chemosensitivity in TMZ treated glioma.

Experimental design

The experimental study design and methodology are appropriate and as per standards.
The statistical analysis is up to the mark. The in vivo work could have been better. The authors compared only two cell lines (relating low grade vs malignant glioma; and chemosensitive vs chemoresistant would added points to the interpretation)

Validity of the findings

The findings need to be validated on different cell lines. Also, validation for off-target effects of Borneol is required. Its role can be tested in metastatic phenotype.

Additional comments

Overall, there are few concerns and the manuscript needs minor revision to recommend for the acceptance.
Review Comments:
• In the abstract, statement 25 may be reframed mentioning the type of rat.
• There are many redundant statements in the introduction and results. The manuscript may be thoroughly rechecked. Sentence in the Line 47 and Line 65 is redundant.
• IC50 studies and dose dependent analysis for TMZ in U251 and C6 cell lines which is discussed under first para of results may be placed in methodology section while retaining the results.
• For Borneol, the dose was taken at 40ug/ml. Details pertaining to IC50 or dose comparison analysis need to be elaborated in the text, although represented in graphs.
• The analysis would have been better to verify in hypoxic condition to confirm the role of Borneol in HIF1 pathway.
• The supporting references and comparative studies on Borneol are missing in the discussion. The discussion section can be revised in a better way projecting the outcome of the study and relating to the novelty of the findings.

---

## Round 0.2 · Minor Revisions

Dear Dr. Cai,

I am happy to let you know that the revised manuscript has been reviewed and the reviewers have provided a favorable recommendation. However, as commented by Reviewer 2, I agree that it is necessary to address these points before the manuscript can be considered further. I welcome you to re-submit the manuscript after addressing points raised by Reviewer 2.

Best,
Himangshu Sonowal

Reviewer 1 ·

Basic reporting

No Comments

Experimental design

No comments

Validity of the findings

No comments

Additional comments

The authors addressed all my comments in a proper way and improved the manuscript in the revised version.

Reviewer 2 ·

Basic reporting

The manuscript has improved, and the authors have made considerable effort to address the comments made in the previous round of review.
- In the revised manuscript, the authors appear to have adjusted the interpretations surrounding the synergistic effect in the results section, but the abstract still emphasizes the synergistic effects. It is recommended to revise accordingly to ensure consistency between the results and abstract.
- For the replicate numbers, please indicate if they are technical replicates or biological replicates/independent experiments.
- Tumor weights and SD rat weights were reported in x ± S. It is recommended to define "x" and “S” in the figure caption.

Experimental design

no comment

Validity of the findings

The authors addressed most of the comments from the last round of review. A few points of concern remain:
- The rebuttal letter indicates that the authors “have added a synergy scoring component for drug combinations”, but these results were not found in the revised manuscript. This is needed if the authors try to justify a synergistic effect. Otherwise, it is recommended to be cautious in the interpretation of the data and avoid suggesting synergy when it is not yet fully justified.
- The drug combination effects were not particularly impressive, according to Figure 1C. It is recommended to ensure that the interpretations are not overstated.

Reviewer 3 ·

Basic reporting

The authors have revised and edited the manuscript thoroughly as suggested.
supporting literature has been added and discussion is also improvised. Additional data required has been furnished. Hence, the current manuscript meets the standards of the Journal and may be recommended for its publication.

Experimental design

The authors mentioned that they could not conduct the experimental studies suggested in their revision in the current condition. Chemosensitive vs chemoresistant comparison would be taken up in subsequent studies. However, other relevant and required data was furnished. The authors further could justify all the comments raised pertaining to study design and technical aspects.

Validity of the findings

The findings have been validated and the revised results have been uploaded. The results and discussion sections have been revised accordingly.

---

## Round 0.3 · accepted · Accept

Dear Dr. Cai,

Thank you for addressing the reviewer's comments and your submission to PeerJ. I am writing to inform you that your manuscript titled - Borneol promotes autophagic degradation of HIF-1α and enhances chemotherapy sensitivity in malignant glioma - has been accepted for publication. Congratulations!

Best,
Himangshu Sonowal
Academic Editor
PeerJ Life & Environment